# Design, Fabrication, and Application of Large-Area Flexible Pressure and Strain Sensor Arrays: A Review

**DOI:** 10.3390/mi16030330

**Published:** 2025-03-12

**Authors:** Xikuan Zhang, Jin Chai, Yongfu Zhan, Danfeng Cui, Xin Wang, Libo Gao

**Affiliations:** 1Key Laboratory of Instrumentation Science and Dynamic Measurement Ministry of Education, North University of China, Taiyuan 030051, China; sz202206222@st.nuc.edu.cn (X.Z.); cuidanfeng@nuc.edu.cn (D.C.); 2College of Chemistry and Chemical Engineering, Xiamen University, Xiamen 361105, China; cj@zehuo.cc; 3Pen-Tung Sah Institute of Micro-Nano Science and Technology, Xiamen University, Xiamen 361102, China; zhanyf@stu.xmu.edu.cn; 4Shenzhen Research Institute of Xiamen University, Xiamen University, Shenzhen 518000, China

**Keywords:** sensor arrays, flexible, fabrication, mechanisms, applications

## Abstract

The rapid development of flexible sensor technology has made flexible sensor arrays a key research area in various applications due to their exceptional flexibility, wearability, and large-area-sensing capabilities. These arrays can precisely monitor physical parameters like pressure and strain in complex environments, making them highly beneficial for sectors such as smart wearables, robotic tactile sensing, health monitoring, and flexible electronics. This paper reviews the fabrication processes, operational principles, and common materials used in flexible sensors, explores the application of different materials, and outlines two conventional preparation methods. It also presents real-world examples of large-area pressure and strain sensor arrays. Fabrication techniques include 3D printing, screen printing, laser etching, magnetron sputtering, and molding, each influencing sensor performance in different ways. Flexible sensors typically operate based on resistive and capacitive mechanisms, with their structural designs (e.g., sandwich and fork-finger) affecting integration, recovery, and processing complexity. The careful selection of materials—especially substrates, electrodes, and sensing materials—is crucial for sensor efficacy. Despite significant progress in design and application, challenges remain, particularly in mass production, wireless integration, real-time data processing, and long-term stability. To improve mass production feasibility, optimizing fabrication processes, reducing material costs, and incorporating automated production lines are essential for scalability and defect reduction. For wireless integration, enhancing energy efficiency through low-power communication protocols and addressing signal interference and stability are critical for seamless operation. Real-time data processing requires innovative solutions such as edge computing and machine learning algorithms, ensuring low-latency, high-accuracy data interpretation while preserving the flexibility of sensor arrays. Finally, ensuring long-term stability and environmental adaptability demands new materials and protective coatings to withstand harsh conditions. Ongoing research and development are crucial to overcoming these challenges, ensuring that flexible sensor arrays meet the needs of diverse applications while remaining cost-effective and reliable.

## 1. Introduction

With the continuous advancement of modern technology, flexible sensor arrays have become a key research focus across multiple disciplines [1,2,3,4,5,6,7,8,9,10]. This growing interest is driven by their exceptional attributes, including superior flexibility, wearability, and large-area-sensing capabilities. These features make flexible sensor arrays particularly effective in dynamic environments [11,12], where conventional sensors often face limitations [13,14,15]. Their ability to conform to complex, variable surfaces offers a significant advantage over rigid sensors, establishing their indispensability in various industrial applications.

Flexible sensor arrays are especially suited for monitoring physical parameters such as pressure [16,17], strain [18,19], and temperature [20]. These sensors are increasingly used in diverse applications, including smart wearables [21,22,23], robotic tactile sensing [24,25,26], health monitoring [27,28,29], and flexible electronics [30,31,32]. For example, in wearable technology, flexible sensors accurately track physiological data such as heart rate and body temperature. In robotics, these arrays enable tactile perception, enhancing environmental interaction. This versatility highlights the transformative potential of flexible sensor arrays across various technological domains.

A key advantage of flexible sensors is their lightweight, adaptable nature. Unlike traditional rigid sensors, which are bulky and inflexible [33,34], flexible sensors are lightweight, pliable, and can conform to surfaces that change shape or move [35,36,37,38,39]. This makes them highly portable and ideal for applications requiring mobility. The demand for portable, versatile technologies is especially prominent in healthcare, where ease of integration and user comfort are crucial. Flexible sensors can be seamlessly incorporated into everyday items, such as clothing or medical devices, without compromising functionality [40,41,42,43,44].

However, as the applications of flexible sensors expand, several challenges have emerged, particularly in the design and fabrication of large-area sensor arrays [45,46,47]. Developing high-performance, large-area flexible sensor arrays has become a critical research focus. These sensors must not only maintain high performance but also ensure stability over extensive areas. Additionally, factors such as material flexibility, durability, and scalability for mass production are essential considerations in the design of these sensors for practical use [48,49,50,51,52].

To achieve optimal functionality in practical applications, large-area flexible sensor arrays must possess several key properties. A major advantage of these arrays is their ability to bend and stretch, allowing them to conform to surfaces with irregular shapes [53,54,55]. This flexibility is essential for their seamless integration into various devices and systems [56,57]. For example, sensor arrays can be used in wearable health monitoring devices that adapt to the contours of the skin or in flexible displays that wrap around curved surfaces [58,59]. The combination of flexibility and scalability significantly broadens their potential applications, enabling them to meet the needs of emerging technologies in ways rigid sensors cannot. Figure 1 provides a comprehensive overview of the research roadmap for large-area flexible sensor arrays.

Large-area flexible sensor arrays typically consist of multiple functional layers [60,61,62,63,64] (Figure 2), including the substrate, spacer, electrode, and sensitive layers. The substrate and encapsulation layers form the structural foundation of the sensor array and are usually made from flexible materials with strong mechanical properties and durability. These layers support the sensor array and allow it to adapt to different surface geometries, ensuring reliable performance in complex environments. The spacer layer, made from insulating materials, isolates the functional layers and maintains electrical insulation [65,66] between the electrode and sensitive layers, as well as between the upper and lower sensitive layers. This isolation prevents signal interference while ensuring proper sensor operation. The electrode layer, a key electrical component, is responsible for signal acquisition and transmission. Made from conductive materials, it detects variations in pressure or strain [67,68,69] and transmits these signals to the electronic processing unit [70]. The sensitive layer, which is the core of the sensor array, can be made from various materials, including resistive [71,72,73], capacitive [74,75], or ionization [76,77] types, each offering distinct advantages for pressure and strain detection. Resistive layers are highly responsive to pressure changes, capacitive layers excel at detecting minute variations, and ionization layers perform exceptionally well under extreme environmental conditions. By combining these materials, flexible sensor arrays provide accurate detection with superior sensitivity and reliability.

This paper offers a comprehensive review of recent advancements in the design, fabrication, and application of large-area flexible sensor arrays. It systematically examines fabrication techniques, operational principles, and commonly used materials in flexible sensor development, focusing on two primary sensor preparation methods. The paper also presents notable case studies of large-area pressure and strain sensor arrays in real-world applications. Finally, it explores the future prospects of flexible sensor arrays in fields such as smart medicine, robotics, and virtual reality, while addressing the challenges and opportunities in these domains.

## 2. Preparation Process of Flexible Sensors

The fabrication process is a key determinant in the development of flexible sensors, driving technological advancements and accelerating sensor capabilities. The choice of fabrication method directly influences sensor performance, manufacturability, and production costs, highlighting the importance of selecting the right technique [78,79,80]. As demand for flexible sensors increases, researchers are focused on optimizing existing fabrication processes to meet stringent performance and stability requirements [81,82].

The choice of fabrication method is closely tied to the material properties used. Different materials have distinct physical characteristics, which influence the most suitable fabrication approach. Common fabrication techniques for flexible sensors include 3D printing [83,84,85], screen printing [86,87], laser etching [88,89,90], magnetron sputtering [91,92], and molding [93,94] (Figure 3). The following discussion examines these methods in detail, focusing on their relationship with material selection.

3D Printing (Figure 3b) is a versatile technique for producing complex three-dimensional structures [83,84,85]. This additive manufacturing process builds objects layer by layer from a digital model, allowing for intricate and customizable designs. While 3D printing offers design flexibility, it is limited by slower production speeds and material-dependent mechanical properties. Materials with low Young’s modulus may compromise sensor rigidity and stability, while those with a higher modulus enhance mechanical performance. A key limitation is the limited material selection, especially for applications requiring high conductivity or temperature resistance.

Screen printing (Figure 3a) is a cost-effective, rapid, and scalable method for mass production [86,87]. It involves transferring ink through a stencil onto a flexible substrate, with ink viscosity and adhesion playing a crucial role. While it allows for efficient production of simple designs, screen printing has lower resolution capabilities and can struggle with pattern fidelity on flexible substrates due to dimensional instability.

Laser etching (Figure 3c) is a precise method for creating high-accuracy patterns and complex geometries [88,89,90]. This subtractive process uses a high-power laser to vaporize or ablate material, enabling micron-scale feature resolution. Laser etching’s effectiveness depends on material properties like thermal conductivity and optical absorption. However, it can cause thermal distortions, especially in flexible substrates, affecting sensor stability and accuracy.

Magnetron sputtering (Figure 3d) is a physical vapor deposition technique used for high-performance flexible sensors [91,92]. It involves generating plasma to sputter target material onto a substrate, creating uniform thin films. The quality of the film depends on the substrate’s mechanical properties, particularly its Young’s modulus. While this method provides excellent control over film thickness and uniformity, it is limited by slow deposition rates and material constraints.

Molding (Figure 3e) is a replication technique that involves creating a master mold, injecting molten material, and cooling it to form the final product [93,94]. It is ideal for high-volume manufacturing, offering excellent production efficiency. Material rheology and mold compatibility are key factors in process optimization, with the base material’s Young’s modulus influencing mold design. However, mold modification can be costly and challenging, making molding better suited for simple sensor designs.

Each fabrication method offers unique advantages and limitations, depending on sensor design requirements, material properties, and production needs (see Table 1). 3D printing excels in customization but is limited by material compatibility and scalability. Screen printing is cost-effective and scalable for simple designs, while laser etching offers high precision for complex patterns but is less scalable and more expensive. Magnetron sputtering is ideal for high-performance sensors but is constrained by material and scalability limitations. Molding is the best option for mass production of geometrically simple sensors but is limited in design resolution and material variety.

Selecting the appropriate fabrication method depends on both process characteristics and material properties. Key parameters like rheological behavior, elastic modulus, and thermal properties directly affect sensor performance and quality. Combining fabrication techniques strategically can lead to the development of high-performance sensors. Ongoing advancements in fabrication methodologies continue to drive innovations in sensor design and technology.

## 3. Mechanism of Flexible Sensors

The operating principle of flexible sensors is based on the electrical, optical, or mechanical properties of their materials [95,96]. When exposed to external stimuli, these sensors undergo physical changes that alter their electrical, optical, or mechanical characteristics. For instance, pressure sensors typically show measurable changes in resistance or capacitance when mechanical pressure is applied.

The structural configuration of flexible sensors significantly impacts their performance. Two primary designs are commonly used: sandwich-type structures [97,98] (Figure 4b) and fork-finger structures [99,100] (Figure 4c). Sandwich-type structures are known for their high integration capability, achieved by vertically stacking multiple functional layers, including conductive, inductive, and insulating components. This design optimizes spatial efficiency and enables seamless integration with other electronic systems. Additionally, sandwich structures tend to have good mechanical recovery, as their multilayer architecture helps restore the original form after stress or deformation. Fabricating sandwich structures requires precise material selection and bonding techniques, often involving advanced manufacturing methods like sputter deposition, thin-film technology, or laser processing. While this increases design and manufacturing complexity, it also enhances the sensor’s performance and stability. In contrast, fork-finger structures have lower integration levels and focus on the geometric configuration of the sensing area. This design often requires multiple contact points and complex electrode arrangements, limiting its use in miniaturized devices. Although simpler to produce and suitable for low-cost sensor production, fork-finger structures may struggle with high-accuracy requirements, especially those demanding high sensitivity and fast response times. They are typically manufactured using photolithography and etching, but their simplicity can result in lower response and stability in high-precision applications.

Flexible sensors operate based on either resistive or capacitive effects, each with its own set of advantages and limitations. The resistive effect involves a change in the sensor’s resistance due to external pressure or deformation [101]. The sensor’s resistance can be expressed by the following Equation (1):(1)R=ρL/A
where ρ is the resistivity, L is the length, and A is the cross-sectional area. In resistive sensors, the material’s resistivity is a key factor in sensitivity and stability. The material’s conductivity, thermal stability, and interaction with the external environment (e.g., humidity and temperature) all influence the sensor’s performance. In environments with high humidity or large temperature variations, resistive sensors may experience significant disturbances, leading to reduced sensitivity or slower response times [102]. In contrast, the capacitive effect involves a change in capacitance to detect pressure or deformation [103], expressed in the Formula (2):(2)$$C=\frac{\varepsilon S}{4\pi kd}$$
where ε is the relative dielectric constant, k is the electrostatic constant, S is the overlap area of the two parallel plates, and d is the distance between the plates. The performance of capacitive sensors is influenced by the dielectric properties of the material, the choice of conducting material, and the state of the electrode surface. In particular, the dielectric constant of the material is crucial for sensitivity. Emerging ionic electronic sensors that use ionic conductors instead of conventional dielectric materials can significantly improve sensitivity. According to the electrical double layer (EDL) theory [104], the inductive capacitance at the EDL interface is given by the formula:(3)$$C_{EDL}=UAC\times A$$
where UAC represents the unit area capacitance, which is primarily influenced by material properties such as ion type and density. UAC can reach values on the order of several μF/cm^2^ due to the nanometer-scale distance between positive and negative charges at the EDL interface. A represents the interconnection area between the ionic layer and the electrodes. Ionic electronic sensors enhance capacitance variation, thereby improving the sensitivity and sensing range of capacitive sensors. To further enhance sensor performance (Figure 4a), Libo Gao et al. [105] utilized MXene as the electrode material in the ionic sensor, resulting in a significant improvement in sensor sensitivity (S_max_ > 45,000 kPa^−1^).

The sensitivity, response time, and stability of sensors can be significantly affected by external environmental conditions. Factors such as temperature and humidity can interfere with the performance of both resistive and capacitive sensors. Temperature changes affect the conductivity and dielectric constant of the material, resulting in changes to the sensor’s electrical characteristics. For resistive sensors, temperature-induced changes in resistance can lead to measurement errors, so temperature compensation mechanisms must be considered in design. For capacitive sensors, temperature changes may also cause capacitance drift, especially in high-temperature environments where changes in electrode spacing and dielectric constant directly impact measurement accuracy. Humidity changes have a more significant effect on resistive sensors, especially when the material has high hygroscopicity. Increased humidity can alter the material’s conductivity, affecting resistance stability. Capacitive sensors, on the other hand, are generally more immune to humidity, but extreme humidity fluctuations can cause electrolytic reactions or other chemical changes on the electrode surface, potentially compromising sensitivity and stability.

For pressure detection applications, flexible sensors are typically categorized into four types: sandwich resistive sensors [106], sandwich capacitive sensors [4], fork-finger resistive sensors [107], and fork-finger capacitive sensors [108] (Figure 4b,c). Sandwich resistive sensors are commonly used in pressure detection due to their structural simplicity and cost-effectiveness. However, their limited sensitivity and response speed may restrict their use in high-precision applications. Sandwich capacitive sensors offer superior noise immunity and can maintain measurement accuracy in complex environments, though they come with increased design complexity and fabrication costs. Fork-finger resistive sensors, with more complex architectures, are particularly suitable for applications requiring rapid response and high precision, thanks to their enhanced sensitivity and response characteristics. However, these advantages come at the cost of greater manufacturing complexity and higher costs. Fork-finger capacitive sensors combine the inherent noise immunity of capacitive sensing with the sensitivity advantages of fork-finger designs, making them ideal for high-precision pressure detection, though they entail significant design and manufacturing expenses.

These sensor configurations (sandwich resistive [109], sandwich capacitive [110], fork-finger resistive [111], and fork-finger capacitive [112,113]) are also widely used in strain detection applications. Sandwich resistive sensors are commonly used for strain measurements due to their high sensitivity and simple design, although their susceptibility to environmental factors can lead to measurement inaccuracies. Sandwich capacitive sensors (Figure 4d) offer excellent noise immunity, making them suitable for complex environments, but their relatively lower sensitivity may limit their effectiveness in high-precision strain measurements. Fork-finger resistive sensors (Figure 4e) offer rapid response times and high sensitivity, making them ideal for accurate strain measurements. However, their implementation is constrained by increased complexity and cost. Fork-finger capacitive sensors combine high sensitivity with robust noise immunity, making them adaptable to various environmental conditions in strain detection, though their practical use is challenged by manufacturing complexity and high costs.

**Figure 4 micromachines-16-00330-f004:**
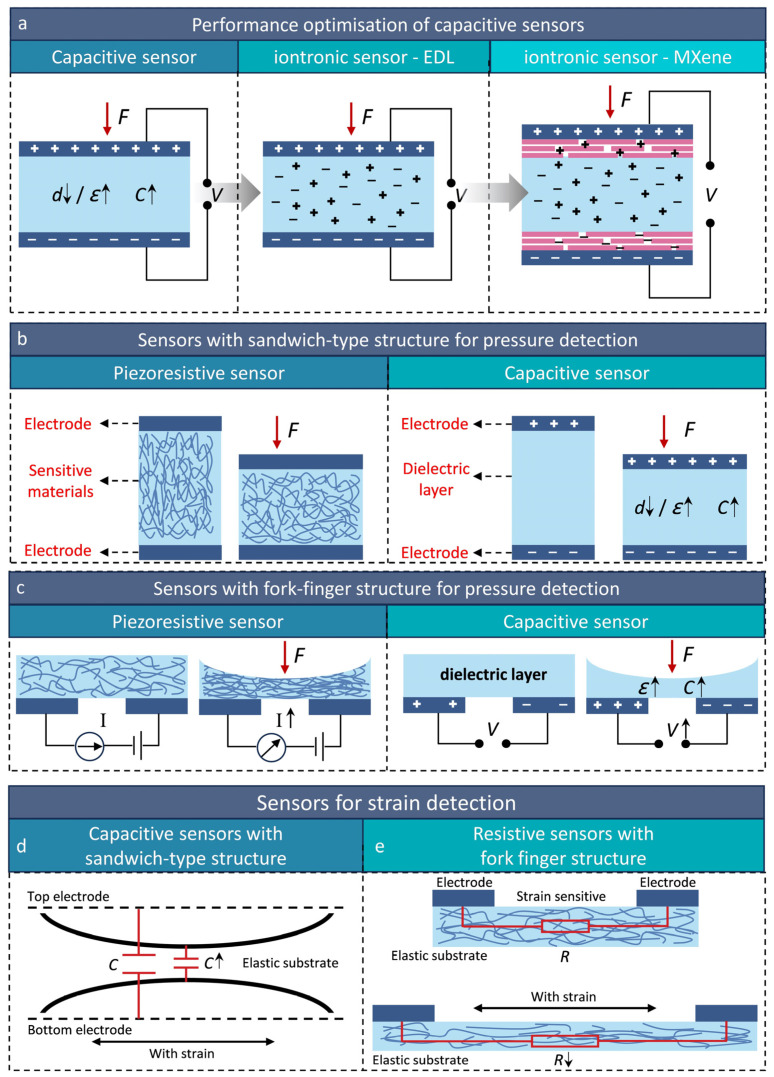
(**a**) Conventional parallel-plate capacitive sensor, ionic sensor based on EDL electrode material, and ionic sensor based on MXene electrode material; reproduced with permission from Springer Nature (2021) [105]. (**b**) Sandwich type resistive sensors and sandwich type capacitive sensors for pressure detection. (**c**) Fork-finger type resistive sensors and fork-finger type capacitive sensors which can detect pressure. (**d**) Strain processes for capacitive sensors of sandwich-type construction. (**e**) Strain processes of resistive strain sensors of the fork-finger type structure.

Advancements in flexible sensor technology have led to increased research on composite materials and multifunctional sensors [114,115]. By combining various materials and sensing mechanisms, researchers are developing flexible sensors with enhanced sensitivity, broader applications, and improved environmental adaptability.

## 4. Common Materials for Flexible Sensors

The selection of materials for flexible sensors is crucial due to their role as innovative electronic devices with diverse applications. These sensors must not only exhibit excellent mechanical properties but also meet requirements for electrical performance, chemical stability, and adaptability to complex environments. Therefore, choosing the right materials is essential for optimizing sensor performance. The fabrication of these sensors depends heavily on the materials used for substrates, packaging, electrodes, and sensitive components. This paper explores the four key characteristics of these materials: flexibility, heat resistance, long-term stability, and interfacial bonding ability.

Substrate and encapsulation materials vary widely, including Polyethylene Terephthalate (PET) [116], Polyimide (PI) [117], Polydimethylsiloxane (PDMS) [7], Polyethylene Naphthalate (PEN) [118], Thermoplastic Polyurethane (TPU) [119], Polyvinyl Alcohol (PVA) [120], Polyethylene (PE) [121], Polyurethane (PU) [122], Polyvinyl Chloride (PVC) [123], Polylactic Acid (PLA) [124], Polyacrylamide (PAM) [125], Polyamide (PA) [126], Fabrics [127], and others (see Table 2). Among these materials, PDMS, TPU, PE, PU, and fabrics offer the best flexibility, particularly in bending and deformation, while maintaining stability throughout these processes. Xin Wang et al. [7] used PDMS to encapsulate flexible sensor arrays, enabling them to conform to the wrist and perform robustly during stretching, bending, twisting, and arching. In contrast, PVC exhibits poor flexibility and is unsuitable for applications requiring significant flexibility. Regarding heat resistance, PI materials excel at maintaining stability at higher temperatures, while PVA, PE, PVC, PLA, and PAM have lower heat resistance and tend to degrade or deform when exposed to heat. As a result, PI is the preferred choice for high-temperature environments. M. Menichelli et al. [117] developed flexible sensors by depositing hydrogenated amorphous silicon on a PI substrate, followed by baking at 100 °C for annealing tests, which significantly improved their X-ray sensitivity.

For long-term stability, materials like PET, PI, PEN, PLA, PAM, and PA perform exceptionally well, retaining their physical properties over extended periods, making them ideal for applications requiring consistent performance over time. Ryosuke Nitta et al. [116] used PET as a substrate to create a flexible humidity sensor, which showed no reduction in humidity sensitivity after 1000 cycles, demonstrating its reliability and durability under mechanical bending. In contrast, some materials may degrade or experience changes in their physical properties over time. For interfacial bonding, materials such as PI, PDMS, TPU, PVA, PU, PLA, PAM, PA, and fabrics exhibit superior bonding abilities, forming strong connections with other materials and ensuring robust adhesion in multi-layer structures. This is particularly important in demanding applications where interfacial bonding is critical. Hui Sun et al. [119] used TPU as a substrate for flexible strain sensors, which were bonded by drying at 25 °C for six hours.

With the increasing use of flexible sensors in biomedical applications, the biocompatibility of materials is crucial. In medical settings, sensor materials must be compatible with human tissue and free from adverse biological reactions. PDMS and PU are commonly used for wearable sensors due to their excellent biocompatibility, allowing them to safely contact human skin for extended periods. Liangsong Huang et al. [122] designed flexible strain sensors using PU film as a substrate, capable of detecting small motion changes when attached to human joints.

Electrode materials commonly used include conductive silver paste [128,129], liquid metal [130,131], copper- and gold-plated materials [132], carbon-based materials [133], PEDOT:PSS [134], and silver nanowires [135] (see Table 3). Liquid metals and silver nanowires offer optimal flexibility and conductivity, with the ability to accommodate large deformations while maintaining high conductivity, making them ideal for flexible electronics. Renchang Zhang et al. [130] designed a flexible capacitive sensor based on liquid metal that retains its metallic conductivity when bent. However, copper- and gold-plated materials provide less flexibility and are generally unsuitable for flexible applications. In terms of thermal resistance, copper-plated, gold-plated, and carbon-based materials perform well, maintaining stability even at high temperatures. Liquid metals and PEDOT:PSS, however, have lower thermal resistance and may degrade when exposed to heat.

For long-term stability, conductive silver paste, copper- and gold-plated materials, carbon-based materials, and silver nanowires retain their conductive properties and stability over time, making them ideal for applications with stringent long-term performance requirements. Wei Zhu et al. [132] designed flexible strain sensors that were cyclically stretched 10,000 times at 25% strain with no significant signal degradation. On the other hand, liquid metals and PEDOT:PSS may experience performance degradation over time. Regarding interfacial bonding, conductive silver paste, PEDOT:PSS, and silver nanowires offer strong bonding, ensuring stable adhesion to the substrate. Chae Young Park et al. [128] used silver paste to create the conductive layer of a flexible sensor that receives electrical signals from the sensitive material. While copper- and gold-plated materials also exhibit good bonding properties, they are more customizable and can be tailored to specific needs.

Sensitive materials commonly used include nanofibers [136,137], polymers [138], conductive fillers [139], organic frameworks [140], hydrogels [141], and metals [142] (see Table 4). Nanofibers, polymers, and hydrogels exhibit excellent flexibility, particularly for flexible sensors, as they can undergo significant deformation while maintaining high sensitivity. For instance, Xinyu Wei et al. [141] developed hydrogel-based flexible sensors that can be attached to the elbow and knee joints to detect various joint movements. However, metals offer less flexibility and are unsuitable for applications requiring highly flexible materials. In terms of heat resistance, metals outperform other materials, maintaining stable performance even at high temperatures. In contrast, organic frameworks and hydrogels have the lowest heat resistance and may degrade or undergo structural changes under high heat.

For long-term stability, nanofibers and metals are superior, with metals particularly retaining electrical conductivity and mechanical strength over time. Bowen Zhu et al. [136] developed a wearable pressure sensor based on gold nanowires, demonstrating stable performance after 10,000 cycle tests. Polymers and conductive fillers offer adjustable long-term stability, which can be tailored to specific application requirements. In terms of interfacial bonding, organic frameworks are the most effective, forming strong bonds with other materials to ensure overall structural stability. Hugo Spieser et al. [140] designed flexible sensors using metal-organic frameworks that can bond stably with graphene-carbon materials. Nanofibers, conductive fillers, and metals also provide adjustable bonding abilities that can be optimized for specific applications.

Material selection is crucial for the performance of flexible sensors. Advances in research have led to the development of various conductive polymers [143], carbon-based materials [144], metals [145], nanomaterials [146], and composites [147], which are now widely used in flexible sensors. As materials science continues to evolve, new materials will emerge, further advancing the development and application of flexible sensor technology.

## 5. Typical Preparation Process of Flexible Sensors

The preparation technology for flexible sensor arrays is critical to enhancing their performance and enabling large-scale applications. Two main preparation methods are commonly used in flexible sensor fabrication: the molding method and the printing method. The molding method is known for its high precision and ability to create complex microstructures [93,94,148], while the printing method stands out for its efficiency, low cost, and potential for large-scale production [7,87]. Below, we explain the characteristics of these methods and their impact on sensor performance through specific case studies.

The molding method enables precise replication of microstructures using prefabricated molds and is capable of producing functional layers with complex geometries and high consistency. For instance, Ningning Bai et al. [148] used this method to create a flexible pressure sensor array with high sensitivity (49.1 kPa^−1^), excellent linearity (R^2^ > 0.995), and a wide pressure response range (up to 485 kPa). The fabrication process involves three key steps (Figure 5a): (1) Fabricating the Structured Ionic Membrane: A semi-ellipsoidal array with upright micropillar structures was created by molding a PVA/H_3_PO_4_ layer to serve as the ionic membrane. This design significantly enhances both sensitivity and pressure response range. (2) Fabricating the Structured Electrodes: Flexible electrodes were made from epoxy/Au materials with uniform microdome structures. The interlocking design of the electrodes and the ionic membrane optimizes their interface, boosting pressure-sensing performance. (3) Device Packaging: The ionic membrane and electrodes were precisely aligned and encapsulated to form a complete sensor array. This sensor demonstrated excellent performance in robotic grasping tasks, achieving over 88% accuracy in object weight recognition using machine learning. These results highlight the advantages of the molding method in producing high-performance sensors and provide a valuable reference for designing and fabricating complex microstructures.

The printing method has become the preferred technology for large-scale production of flexible sensors due to its high efficiency, low cost, and scalability. Xin Wang et al. [7] used the printing method to create a fully printed flexible pressure sensor array (6 × 9) with high sensitivity across a wide linear range from 4.7 to 100 kPa (3.997 kPa^−1^, R^2^ = 0.999). The fabrication process of the printed sensor is divided into three main stages (Figure 5b): (1) Layers of electrodes, insulation, sensitive materials, microstructures, and spacers are printed sequentially. The precision of the printing and the choice of materials at each stage are crucial for the sensor’s final performance. (2) Bottom Sensor Preparation: The same printing process is applied to the bottom layer to ensure consistency and compatibility with the top layer. (3) Sensor Packaging: The top and bottom sensors are aligned and encapsulated with PDMS to protect the internal structure and enhance mechanical stability. The printing method’s efficiency and scalability make it ideal for the low-cost production of large-area sensor arrays. Performance consistency can be further improved by optimizing printing materials and process parameters.

Both the molding and printing methods offer unique advantages and are suited to different applications. The molding method is ideal for creating high-precision, complex microstructures, while the printing method is better suited for large-scale, cost-effective production. In the future, integrating both technologies may become a key development direction. Combining the molding method’s precision with the printing method’s efficiency could lead to a hybrid process that improves sensor performance while reducing manufacturing costs. As these technologies evolve, flexible sensors will not only offer higher performance but also enable the widespread use of large-area sensor arrays in fields such as smart healthcare, robotics, and wearable devices.

## 6. Typical Applications of Large-Area Pressure-Based Sensor Arrays

Advancements in flexible sensing technology have rapidly transformed large-area pressure sensor arrays from laboratory concepts into versatile platforms for a wide range of applications. With skin-like flexibility, millimeter-level spatial resolution, and ultra-high-pressure sensitivity, these devices are revolutionizing human–computer interaction, enabling intelligent systems to detect touch digitally. Currently, applications are evolving in three main directions: high-sensitivity pressure mapping systems [149], deformation-resistant dynamic monitoring platforms [150,151], and multimodal integrated sensing interfaces [152,153]. These innovations are driving significant changes in healthcare, industrial robotics, and smart wearables.

In fields like medical diagnostics and human–computer interaction, the ability to detect subtle pressure changes is pushing sensor sensitivity to new limits. Zekun Yang et al.’s [154] development of h-BN-enhanced ionic pressure sensor arrays (15 × 15) offers innovative solutions for these applications (Figure 6a). The sandwich structure, created using screen printing, combines h-BN-doped ionic ink with silver paste electrodes, achieving ultra-high sensitivity of 261.4 kPa^−1^ and a broad response range from 0.05 to 450 kPa. This microstructural innovation enables the sensor to maintain linear response at high pressures (up to 400 kPa), overcoming the limitations of traditional devices with restricted dynamic ranges. The value of this innovation lies not only in its performance but also in its potential for large-scale manufacturing using printed electronics. The team also developed embedded signal processing circuits that convert raw signals into biomechanical parameters, constructing 3D images of wrist pressure in real time to monitor the distribution of forces on wrist bones. This provides a new tool for preventing tendon sheath cysts and enhancing wrist healthcare (Figure 6b). However, the printing process’s linewidth limitation (>500 µm) challenges the reduction of the spacing between adjacent sensing units, making array density and signal crosstalk significant obstacles to high-resolution pressure mapping.

To address this issue, Junli Shi et al. [155] advanced high-sensitivity pressure mapping systems by innovating micro-nano-processing technology. They used laser etching to perforate holes in the PDMS membrane, creating a hole array, and injected ion gel precursor into the holes to form the ion sensing element array (28 × 28). Embedded electrodes were created via electron-beam evaporation and spin-coating methods. The resulting sandwich-type capacitive pressure sensor array combined the two electrode layers with the ion sensor array layer (Figure 7a). The sensor array cross-section is shown in Figure 7b. This study achieved high sensitivity (>174 kPa^−1^ from 0.15 Pa to 400 kPa) while minimizing signal crosstalk. The sensor array successfully enabled dynamic pressure mapping for single-finger, two-finger, and hammer taps (Figure 7c). In a robotic haptic application, the sensor array was successfully attached to the palm of a prosthetic hand, monitoring the grasping of 10 different objects (Figure 7d), including plastic balls, dolls, batteries, triangular blocks, adhesive tapes, glass bottles, tweezers, mice, scissors, and a 500 g weight. Combined with a one-dimensional convolutional neural network (1D-CNN) model, the sensor array achieved an average classification accuracy of 99.5% (Figure 7e). These results demonstrate not only high sensitivity and mechanical robustness, but also provide a new technological path for high-precision detection of complex pressure distributions.

For wearable devices or soft robots, sensors must be able to withstand complex mechanical deformations. Yangbo Yuan et al. [156] designed an 8 × 8 resistive array that offers a solution to this challenge (Figure 8a). Innovations include: (1) a polyimide microporous isolation layer (PIL) that maintains sensitivity of >21.5 kPa^−1^ even with 256 m^−1^ curvature bending, thanks to a stress dissipation mechanism; (2) a serpentine copper electrode topology that provides 45% tensile deformation tolerance, essential for human–robot collaboration. Experiments show that the sensor can accurately detect finger pressure during dynamic bending, addressing the industry challenge of “motion artifacts” in flexible electronics (Figure 8b). Without the PIL, the sensor is strongly interfered with when the soft gripper bends and stretches during the gripping process (Figure 8c). More impressively, the system integration strategy combines an adaptive filtering algorithm with the robot control system, enabling real-time closed-loop feedback of the deformation surface pressure field, crucial for the tactile feedback of intelligent prosthetic limbs. However, monitoring human health on curved surfaces requires not only improving the sensor arrays’ ability to withstand complex deformations but also increasing their density to allow for conformal attachment to the human body and real-time physiological signal detection.

To address this challenge, Xin Wang et al. [7] developed a high-density pressure sensor array (6 × 9) with high sensitivity (3.997 kPa^−1^) and a wide linear range from 4.7 to 100 kPa (R^2^ = 0.999). The array’s robust performance was confirmed through finite element analysis under tension, bending, torsion, and twisting. It can conformally attach to the forearm, acquire real-time radial artery pulse data at the wrist, and reconstruct the radial artery’s width and length without requiring precise positioning (Figure 9a). When combined with machine learning algorithms, the system enables accurate blood pressure monitoring (Figure 9b). This study overcomes the limitations of traditional sensor arrays in flexible attachment and high-precision measurement, demonstrating the potential of high-density sensor arrays for biomedical monitoring. The accurate capture of wrist pulse data allows for real-time health monitoring, which is especially useful for reconstructing pulse and arterial morphology. It also supports the development of wearable smart medical devices.

As human–computer interaction evolves, a single pressure-sensing dimension is no longer sufficient for complex environmental sensing. e-Skin, developed by Yogeenth Kumaresan et al. [157], represents a significant advancement in sensing capabilities (Figure 9c). This device integrates 4 × 4 temperature sensing units on top of an 8 × 8 pressure array, providing fast response (2.5 s) and recovery (4.8 s) times through CNT/PEDOT:PSS composites. Key innovations include: (1) a hardness-gradient PDMS dielectric layer enabling a wide-area pressure response (0–10 kPa), and (2) a spatio-temporal decoupling algorithm that effectively differentiates pressure and temperature signals. The circuit reads pressure changes and generates a heat map, with the red square indicating the touch position. To test its ability to distinguish between pressure and temperature, three areas are tested: Area A responds to pressure, Area B to heat, and Area C to increased heat, demonstrating the array’s ability to selectively differentiate pressure and temperature. This multi-physical sensing capability allows the device to capture both contact force and surface heat flow, mimicking the mechanical-thermal sensing function of human skin.

Large-area flexible pressure sensor arrays have numerous potential applications in healthcare, robotics, and wearable devices. Their key performance indicators include array size, detection range, minimum detection limit, response time, and durability (see Table 5 [7,29,150,154,155,156,157,158,159,160,161,162]). Current sensor arrays exhibit several limitations: (1) Some sensors have a narrow detection range (0–10 kPa), limiting their broader application; (2) sensitivity varies greatly between sensors, with some offering only 0.19 kPa^−1^, which is insufficient for high-precision measurements; (3) response times for some sensors are long (e.g., 2.5 s), hindering real-time monitoring; (4) durability is limited, with some sensors cycling only 200 times, making them unsuitable for long-term use. Among the sensors listed, the PBU and AgNW sensor stands out with the best overall performance, featuring a sensitivity of 888.79 kPa^−1^, detection range of 1–100 kPa, minimum detection limit of 0.4608 Pa, response time of 66 ms, and a durability of 10,000 cycles. This sensor excels in high-precision pressure detection and stable long-term operation, making it ideal for medical and robotics applications. In contrast, sensors made with PDMS, TPU, and conductive ink have lower sensitivity (10.50 kPa^−1^), but their larger array size (64 × 64) and durability (up to 45,000 cycles) make them well-suited for applications requiring large-area coverage and high durability. Overall, there is significant room for improvement in terms of sensitivity, detection range, and durability, and future research should focus on optimizing these parameters to expand their applications.

Despite breakthroughs in many areas, large-area pressure-based sensor arrays still face significant challenges, including balancing high sensitivity with a wide range, addressing mechanical flexibility and electrical stability, and tackling the complexity of multimodal integration and signal decoupling. Current research has partially addressed these challenges through microstructure engineering [163,164] (e.g., gradient stiffness design), interface optimization [65] (e.g., stress-dissipating isolation layers), and intelligent algorithms [165,166] (e.g., spatio-temporal signal separation). However, further progress is needed in areas such as long-term reliability in dynamic environments (>10^6^ cycles), biocompatible packaging, and large-scale fabrication of sensor arrays (>100×100). Notably, flexible sensing technology is shifting from detecting single physical quantities to multi-dimensional coupled sensing, which places greater demands on the material systems and architectures of these devices.

## 7. Common Applications of Large-Area Strain Sensor Arrays

The limitations of traditional pressure sensing technologies have led to the development of cross-scale sensing concepts. The planar design of conventional pressure sensor arrays struggles to meet the needs of reconstructing complex deformation fields, especially when capturing both normal pressure and tangential strain simultaneously. This gap has prompted researchers to explore strain-pressure synergistic sensing systems, where large-area strain sensor arrays compensate for the limitations of pressure sensors by monitoring mechanical behaviors such as surface stretching, bending, and twisting [167]. For example, in wearable health monitoring, strain sensors can accurately track joint movement, while pressure sensors assess contact stress distribution on the body’s surface [168]. Together, these sensors create a more comprehensive biomechanical model.

Flexible strain sensing technology is pushing the boundaries of traditional mechanical testing, transitioning from static deformation analysis to dynamic multi-physical field reconstruction. Large-area strain sensor arrays are becoming the “digital skin” of intelligent systems, allowing them to perceive the physical world thanks to their high micro-strain resolution (<0.1%) and multi-dimensional coupled sensing capabilities. The integration of flexible displays and strain sensing is revolutionizing human-computer interaction. For instance, a 4 × 9 strain sensing array developed by Ming Li et al. [19] addresses the mechanical mismatch problem in flexible electronic devices through innovative interface engineering (Figure 10a): (1) The CB/PDMS/Ecoflex gradient interconnecting layer design ensures a modulus match (Young’s modulus difference < 0.5 MPa) between the silver nanowire-based sensitive layer and the substrate, reducing stress concentration caused by bending; (2) the asymmetric electrode layout optimizes strain transfer efficiency, maintaining stable device performance after 3000 bending cycles. This technology’s core value lies in creating a ‘sensing-display’ closed-loop system: when the user performs different bending gestures, the AMOLED screen provides real-time thermodynamic feedback. This “mechanical visualization” interaction mode has been successfully applied in edge-touch scenarios on folding screen mobile phones, offering a new dimension of interaction beyond capacitive touch.

In structural health monitoring, accurately capturing the strain field across the entire domain is critical for failure prediction [169]. The 30×30 spiked carbon nano-sphere (SCN) arrays developed by Shuxing Mei et al. [170] represent a major advancement in strain sensing accuracy (Figure 10b): (1) SCN arrays, formed by a self-assembly process, generate the Fowler–Nordheim quantum tunneling effect, achieving the GHG effect over 60% of the strain range; (2) these arrays offer ultra-high sensitivity (70,000 GF value), three orders of magnitude higher than traditional carbon-based sensors, and a strain-crack extension correlation model that captures the strain gradient at the crack tip with a spatial resolution of 100 pixels/cm². The device’s disruptive nature lies in the synergistic optimization of material, structure, and algorithm. Simulation experiments on surfaces such as hardwood, soft sponges, and biological tissues showed no signal change when tapped on hardwood, a change on the soft sponge, and corresponding changes when biological tissue was deformed, confirming the sensor’s reliability in biomechanical monitoring. This research enhances device performance and safety, making it ideal for large-scale structural monitoring and wearable devices with complex shapes.

**Figure 10 micromachines-16-00330-f010:**
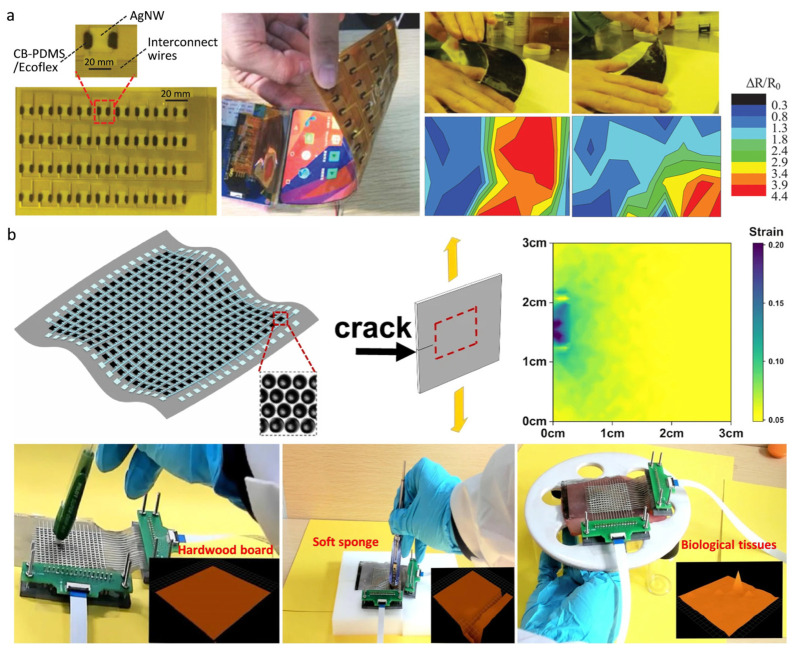
(**a**) A photo image of a 4 × 9 strain sensor array, created by laminating the sensor array onto the back of a flexible AMOLED display panel using optical clear adhesive (OCA) and PI tape to create a flexible interactive surface. The measurement mapping (Δ R/R 0) of the response at each sensor on the array under different bending modes represents the strain distribution; reproduced with permission from Wiley-VCH (2021) [19]. (**b**) The sensing membrane is composed of ordered SCNs arranged in a PDMS matrix and is used to detect the strain distribution near the crack tip. When connected to a hardwood board, the sensing membrane does not respond to poking. When the sensing membrane is attached to the soft sponge, the sensing membrane responds to the poke. When the sensing film is attached to the pork slices, it can detect the deformation of the pork slices in real time; reproduced with permission from Springer Nature (2021) [170].

The demand for mechanical sensing in complex working environments has led to the development of a new strain-pressure decoupled detection paradigm [171,172]. The silica gel-based dual-mode sensor (Figure 11a) developed by Ryosuke Matsuda et al. [167] addresses the challenge of multi-physics crosstalk through material innovation. By designing a carbon nanoparticle/silica composite system, this sensor can independently detect pressure (1–800 kPa) and strain (>50%) over a wide deformation range. Compared to the designs by Ming Li and Shuxing Mei, this sensor significantly enhances performance when pressure and strain coexist, eliminating signal interference. In its application, independent pressure and strain detection (in both the x and y directions) is achieved using a single-pixel device (Figure 11b). When strained along the x and y directions, the resistance increases along those axes, while the application of pressure decreases the resistance. This sensor is effective in real-time monitoring of human movement and physiological states, providing more accurate and stable feedback. Overall, this study introduces new concepts for developing multi-physics sensors with considerable innovative potential and application value.

In conclusion, the wide range of applications for large-area strain sensor arrays across various fields demonstrates their excellent performance and potential. From smart wearable devices to robotics, structural health monitoring, biomedicine, and smart materials, the sensitivity and high resolution of these sensor arrays offer accurate, real-time data for multiple industries. As technology advances, strain sensor arrays are expected to play a key role in emerging fields and drive innovations in related technologies.

## 8. Conclusions and Perspectives

This paper provides an overview of the design, fabrication processes, operating mechanisms, and common materials used in large-area flexible sensor arrays. It also discusses the progress and challenges these sensors face in real-world applications. Flexible sensors are increasingly utilized in diverse fields, including smart healthcare, robotics, and virtual reality, due to their exceptional flexibility, wearability, and adaptability to complex environments. This review covers the preparation methods, working principles, structural features, material selection, and application examples of flexible sensors, highlighting significant advancements in the field.

The paper thoroughly examines key fabrication processes such as 3D printing, screen printing, laser etching, magnetron sputtering, and molding, analyzing the advantages and disadvantages of each method. It emphasizes the close relationship between the fabrication process and material properties, highlighting the importance of selecting the appropriate process to achieve high-performance sensors. Regarding operating mechanisms, the paper introduces the two main principles—resistance and capacitance effects—and explores the operating principles of sandwich-type and fork-finger structures. It also discusses the advantages and limitations of each sensor type in different application contexts.

Material selection is crucial to the performance of flexible sensors. This review examines common substrate, electrode, and sensitive materials, highlighting key performance indicators such as flexibility, heat resistance, and stability. It emphasizes that different materials are optimal for specific applications. The paper concludes with case studies of large-area, pressure-based sensor arrays successfully applied in medical and healthcare, industrial robotics, and smart wearable devices, showcasing their market potential and technical advantages.

Overall, advancements in technology have significantly enhanced the performance and practicality of large-area flexible sensor arrays. These sensors are expected to play an increasingly important role in sectors like smart healthcare, robotics, and virtual reality. However, several challenges remain, including cost-effective mass production, long-term stability, and the integration of multiple functions within sensor arrays.(1)Mass Production Feasibility


Large-scale production of flexible sensor arrays presents a significant challenge due to the complexity of fabrication processes, material costs, and the need for precision. While techniques like 3D printing and screen printing have made progress, further optimization is necessary to achieve high throughput without compromising sensor quality. Streamlining manufacturing steps and minimizing defects will be key to ensuring cost-effectiveness. To improve scalability, automated production lines and advanced robotic assembly systems should be considered. Simplifying the manufacturing process, such as through efficient molding techniques, can also increase production speed and reduce per-unit costs. Crucially, the manufacturing process must remain feasible for large-scale production without compromising sensor performance.(2)Wireless Integration


With the growing demand for real-time data collection and analysis, wireless integration is essential. However, integrating wireless communication systems into large-area flexible sensor arrays presents challenges related to energy consumption, signal stability, and interference management. Long-term wireless operation requires energy-efficient designs, including low-power communication protocols (e.g., Bluetooth Low Energy or Zigbee) and energy harvesting technologies. Ensuring reliable wireless communication in complex environments requires robust shielding techniques and advanced signal processing to mitigate noise and interference. Developing compact, lightweight, and flexible wireless modules is also crucial to integrating them seamlessly into sensor arrays without compromising flexibility and performance.(3)Real-Time Data Processing

Real-time data processing is critical for many flexible sensor array applications, particularly in healthcare and robotics. The challenge is efficiently processing large volumes of sensor data while maintaining low latency and high accuracy. Distributed computing architectures and edge computing approaches must be explored to allow local, real-time data processing, reducing reliance on constant communication with a central server. Additionally, integrating machine learning algorithms to enhance sensor data interpretation will be vital for real-time decision-making. However, incorporating these technologies into flexible sensor arrays without compromising design and stability presents a significant challenge.(4)Long-Term Stability and Environmental Adaptability

To ensure the success of large-area flexible sensor arrays in real-world applications, long-term stability and environmental adaptability are essential. Sensors must endure harsh conditions, including extreme temperatures, humidity, mechanical stress, and exposure to chemicals. Developing materials resistant to environmental degradation and improving structural designs to prevent long-term failure are crucial. For example, using composite materials that combine conductive polymers and nanomaterials can improve both sensitivity and environmental durability. Innovations in encapsulation and protective coatings will also enhance the longevity and reliability of sensor arrays, making them more suitable for deployment across various industries.

## Figures and Tables

**Figure 1 micromachines-16-00330-f001:**
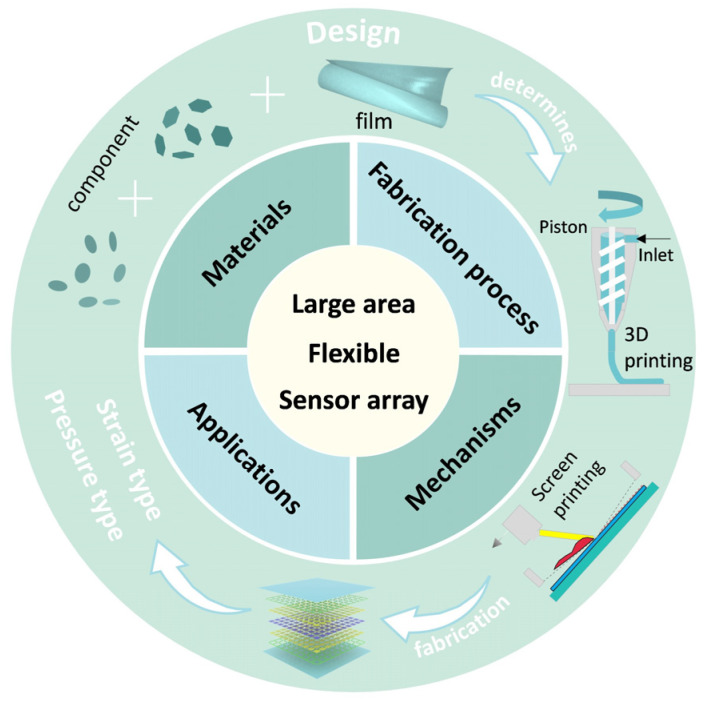
The research route for large-area flexible sensor arrays includes material selection, sensor manufacturing, sensor mechanisms, and applications.

**Figure 2 micromachines-16-00330-f002:**
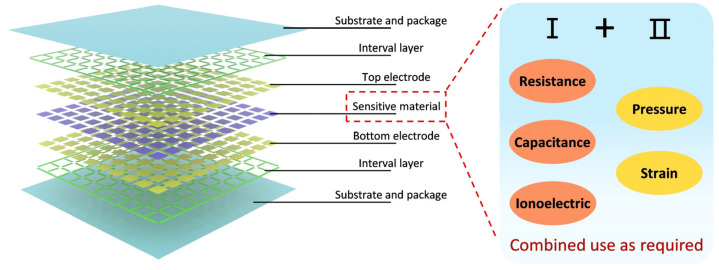
Explosion diagram of large-area flexible sensor array.

**Figure 3 micromachines-16-00330-f003:**
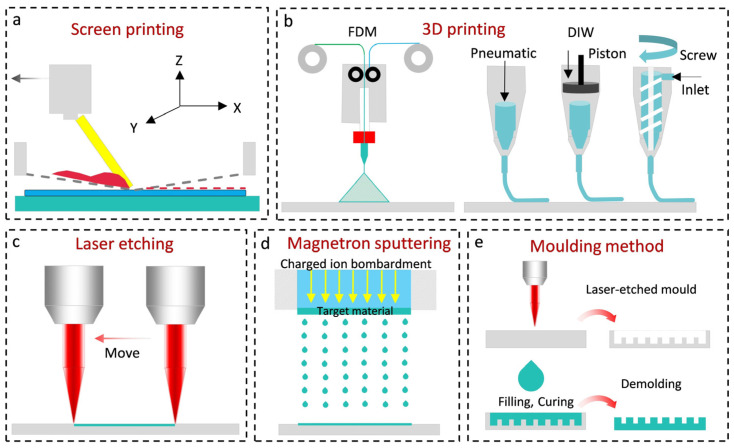
(**a**) Screen printing process. (**b**) 3D printing technology. (**c**) Laser etching process. (**d**) Magnetron sputtering process. (**e**) Molding method.

**Figure 5 micromachines-16-00330-f005:**
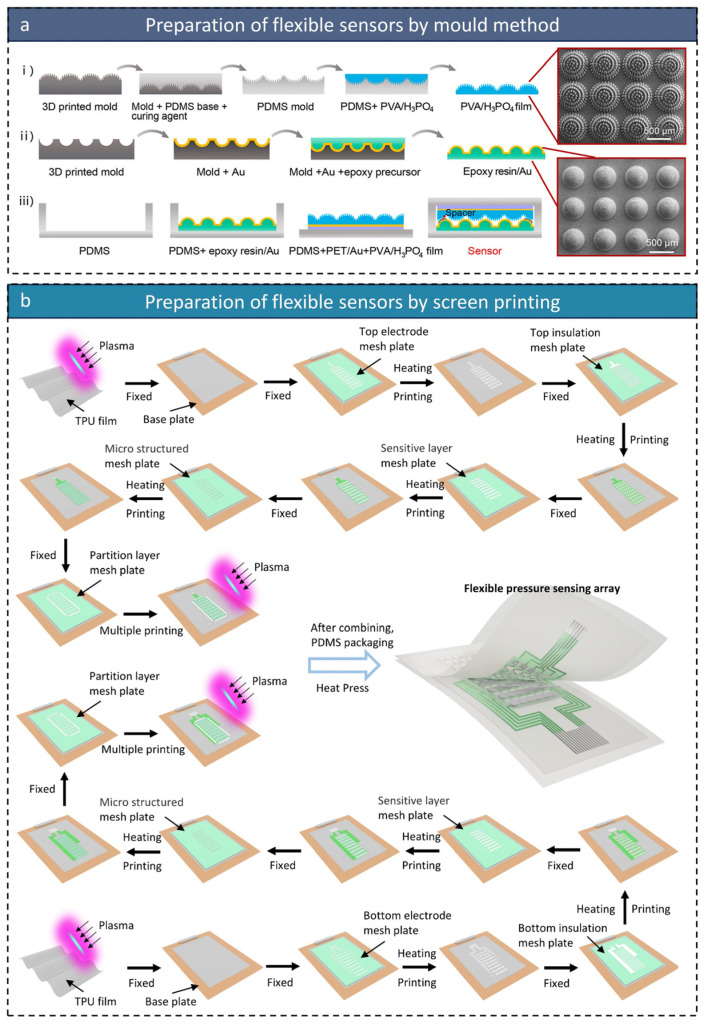
(**a**) Three key steps in fabricating a hierarchical interlocked ionic-electronic pressure sensor using the molding method: (i) fabrication of structured ionic membranes, (ii) fabrication of structured electrodes, and (iii) device encapsulation. SEM images of the top view of the PVA/H_3_PO_4_ membrane and the top view of the epoxy/Au electrode; reproduced with permission from American Chemical Society (2022) [148]. (**b**) Preparation process of fully printed flexible pressure sensor arrays using the printing method; reproduced with permission from Wiley-VCH (2025) [7].

**Figure 6 micromachines-16-00330-f006:**
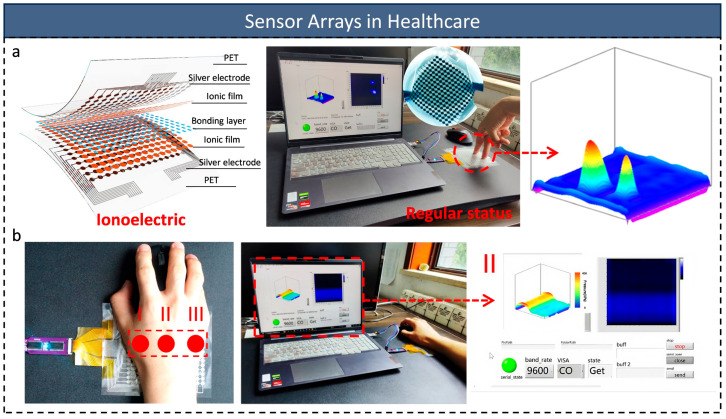
(**a**) Schematic diagram of the Ionic Pressure Sensors (FIPS). After establishing the sensor system, the three-dimensional spatial pressure mapping responds to dual finger pressure. (**b**) The bones of the human wrist attached to the palm of the hand are divided into three parts: the navicular bone (I), the lunate bone (II), and the pisiform bone (III). A sensor array is used to monitor the pressure on the wrist and display a 3D pressure map in real time; reproduced with permission from Springer Nature (2023) [154].

**Figure 7 micromachines-16-00330-f007:**
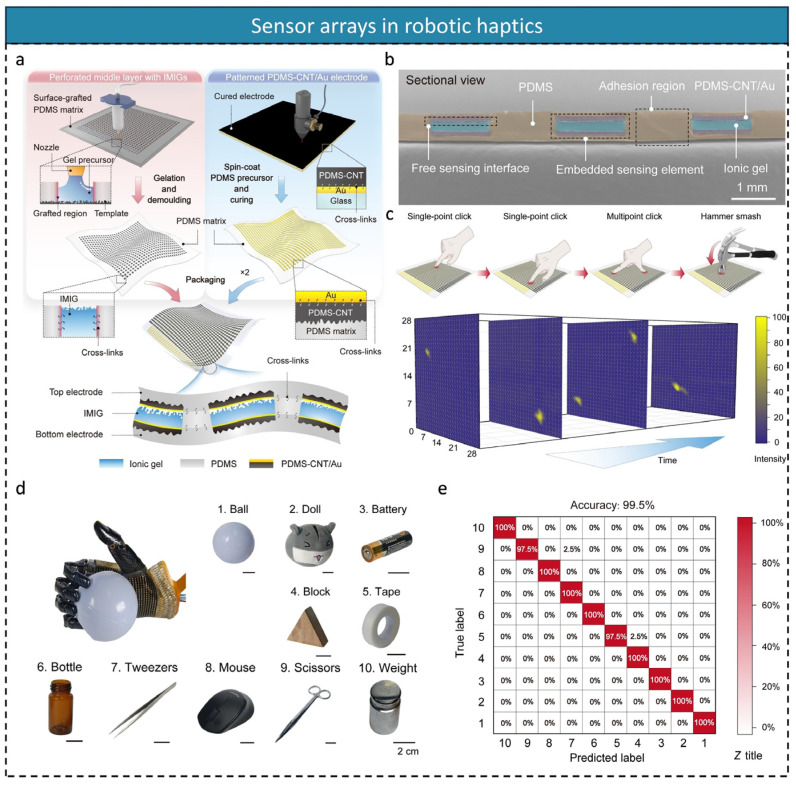
(**a**) Fabrication of pressure sensor arrays. The left column shows the fabrication of a perforated PDMS interlayer with embedded IMIGs; the right column shows the fabrication of embedded electrodes. Two electrode layers and a PDMS interlayer with embedded IMIGs in the middle were combined to integrate into the skin. (**b**) SEM image of the cross-sectional view of the sensor array. (**c**) Schematic of dynamic stimuli and dynamic signal mapping of dynamic processes including single touch, multi-touch, and hammering. (**d**) Recognition of 10 different objects using a sensor array combined with machine learning. (**e**) Confusion matrix of 10 test objects with 99.5% accuracy; reproduced with permission from American Association for the Advancement of Science (2023) [155].

**Figure 8 micromachines-16-00330-f008:**
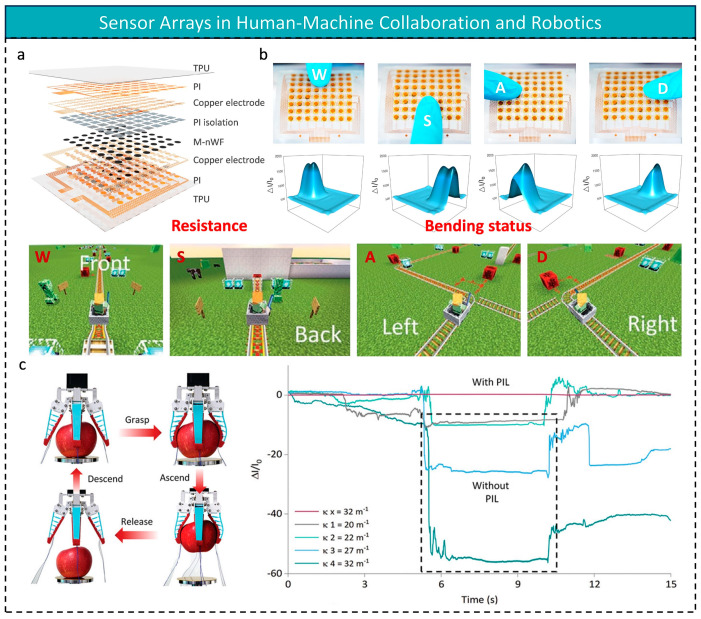
(**a**) Decomposition diagram of an integrated piezoresistive sensor array composed of soft thermoplastic polyurethane encapsulation, PI/Cu serpentine electrodes, PI isolation layer (PIL), and MXene-coated non-woven fabric (M-nWF) sensitive layer. (**b**) Optical images and 3D pressure maps generated by pressing a finger onto a sensor array on a curved surface and using the signals to control the game character to move forwards, backwards, left and right. (**c**) Gripping and transporting apples using a soft gripper with integrated sensors and comparing sensor current changes with and without PIL; reproduced with permission from Wiley-VCH (2024) [156].

**Figure 9 micromachines-16-00330-f009:**
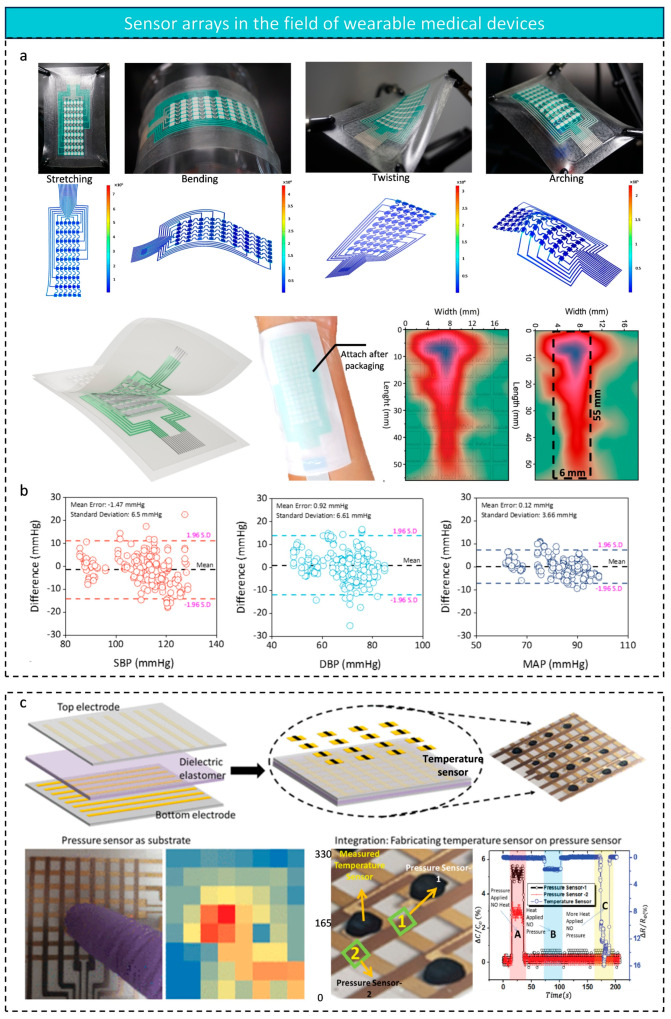
(**a**) Optical pictures of the sensor array in tension, bending, twisting and torsion and its finite element analysis. Schematic of the printing process of the sensor. Schematic of the sensor array conformally attached to the forearm. 2D and 3D pulse mapping from the sensor array. Representative ‘pulse width’ and ‘pulse length’ of the radial artery measured in volunteers. (**b**) Bland–Altman plots of SBP, DBP, and MAP; reproduced with permission from Wiley-VCH (2025) [7]. (**c**) Integration of temperature and pressure sensor layers; optical images and thermal maps when touching the pressure sensing array; reference pixels used during the characterization of integrated temperature and pressure sensing arrays; integrated array response to temperature and pressure; reproduced with permission from Institute of Electrical and Electronics Engineers (2021) [157].

**Figure 11 micromachines-16-00330-f011:**
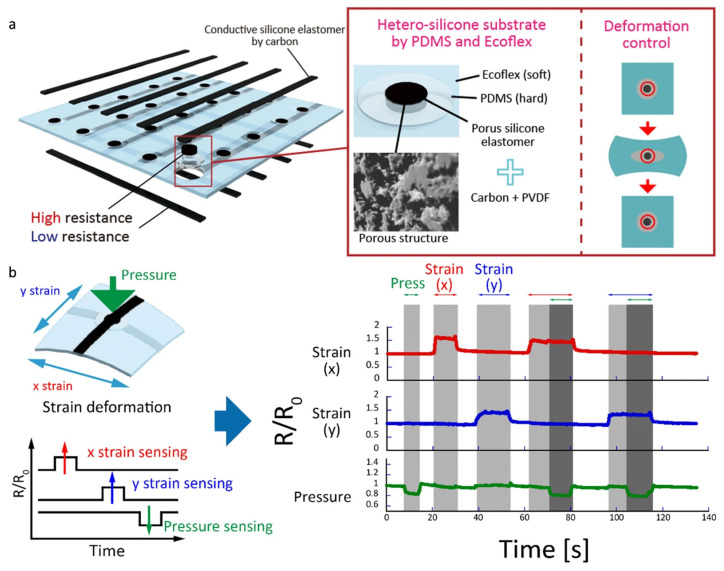
(**a**) Schematic diagram of the array. The silicone substrate is made of two different hardness values of silicone. The harder silicone PDMS can suppress the deformation of pressure-sensing elements during the stretching process. (**b**) Demonstration of independent pressure and x and y strain detection using a single-pixel device. When the device strains in the x and y directions, the resistance along the x and y axes increases. On the contrary, when the sensor of the device is pressed, the resistance of the pressure sensor decreases. In this demonstration, pressure and strain stimuli are perceived independently, while stimuli are applied simultaneously; reproduced with permission from Springer Nature (2020) [167].

**Table 1 micromachines-16-00330-t001:** Advantages, challenges, applicability, cost-effectiveness, scalability, sensor resolution, and material compatibility of common preparation processes for flexible sensors.

Manufacturing Technique	Advantages	Challenges	Suitability for Applications	Cost-Effectiveness	Scalability	Sensor Resolution	Material Compatibility
**3D Printing**	Complex structures can be manufactured, customised designs, high design freedom.	Slow production speeds, limited choice of materials and restricted mechanical properties of the end product, especially with low modulus materials can lead to performance problems.	Suitable for complex, customised designs, especially in low volume production and sensor applications requiring a high degree of design freedom.	Moderate to high; equipment and materials can be expensive, but small batches and custom designs may reduce cost.	Low to moderate; not ideal for mass production due to slow speed and material limitations	Moderate to high resolution, but dependent on material and printing quality. Can create fine details but may lack high precision.	Limited to specific materials (e.g., thermoplastics, some conductive inks); challenges with high-conductivity or high-temperature materials.
**Screen Printing**	Cost-effective, fast production, mass-producible, suitable for relatively simple designs.	Lower resolution, not suitable for complex designs, susceptible to substrate instability leading to pattern misalignment.	Suitable for mass production and low-cost sensor fabrication, it is commonly used in applications such as electronic labels and touch screens.	Low; one of the most cost-effective methods, especially for large-volume production.	High; ideal for mass production with low to moderate design complexity.	Low resolution, suitable for less intricate patterns.	Compatible with a wide range of inks (conductive, non-conductive); limited by the substrate’s flexibility.
**Laser Etching**	High precision, suitable for complex patterns, fine machining of micron-sized structures.	May lead to thermal deformation of the material, susceptible to thermal deformation, especially with flexible substrates.	Suitable for high-precision and complex geometry sensor fabrication, especially for applications requiring micron-level accuracy.	Moderate; setup costs for laser systems are high, but precision improves performance and reduces errors.	Moderate; slower process limits scalability for very large production volumes.	Very high resolution (micron-level); excellent for fine and precise features.	Primarily works with metals, thin films, and some polymers; material type and thickness can affect results.
**Sputtering**	High precision control of film thickness and uniformity, suitable for high performance applications, stable performance especially in harsh environments.	Slower deposition speeds and limited choice of materials, especially flexible substrates may affect film adhesion.	Suitable for high-performance thin-film sensors, such as high-temperature-resistant and highly conductive sensors, which are widely used in thin-film electronics and optoelectronic devices.	Moderate to high; equipment is costly, but performance and longevity of sensors can justify investment.	Low to moderate; slower deposition rate and material limitations hinder large-scale manufacturing.	High resolution for thin films and uniform deposition.	Compatible with a variety of materials, including metals, alloys, and some dielectrics; material flexibility and film adhesion are critical.
**Molding**	High productivity, suitable for mass production, simple process.	Limited by material rheology and mould design, hard materials may affect mould stability.	Suitable for high volume production, commonly used in sensor fabrication on plastic or polymer substrates, such as flexible touchscreens and wearable devices.	High; particularly cost-effective for large batches of simple sensor designs.	Very high; ideal for high-volume production with relatively simple designs.	Moderate resolution; limited by mold design and material flow properties.	Limited by material flow characteristics; works well with thermoplastics and elastomers, but difficult with complex, multilayered materials.

**Table 2 micromachines-16-00330-t002:** Common materials and performance analysis of substrate and package materials.

	Materials	Substrate and Package Materials												
Performances		PET	PI	PDMS	PEN	TPU	PVA	PE	PU	PVC	PLA	PAM	PA	Fabrics
**Flexible**		√	√	√	√	√	√	√	√	ⅹ	√	√	√	√
**Heat-resistant**		√	√	√	√	√	ⅹ	ⅹ	√	ⅹ	ⅹ	ⅹ	√	√
**Long-term stability**		√	√	√	√	√	√	√	√	√	√	√	√	√
**Interface bonding**		√	√	√	√	√	√	√	√	√	√	√	√	√

**Table 3 micromachines-16-00330-t003:** Common materials and performance analysis of electrode materials.

	Materials	Electrode Materials					
Performances		Conductive Silver Paste	Liquid Metals	Copper/Gold Plated	Carbon Based	PEDOT:PSS	Silver Nanowires
**Flexible**		√	√	ⅹ	√	√	√
**Heat-resistant**		√	ⅹ	√	√	ⅹ	√
**Long-term stability**		√	√	√	√	√	√
**Interface bonding**		√	√	√	√	√	√

**Table 4 micromachines-16-00330-t004:** Common materials and performance analysis of sensitive materials.

	Materials	Sensitive Materials					
Performances		Nanofibres	Polymer	Conductive Fillers	Organic Frameworks	Hydrogel	Metals
**Flexible**		√	√	√	√	√	ⅹ
**Heat-resistant**		√	√	√	ⅹ	ⅹ	√
**Long-term stability**		√	√	√	√	√	√
**Interface bonding**		√	√	√	√	√	√

**Table 5 micromachines-16-00330-t005:** Performance of large area pressure sensor arrays.

Materials	Array Size	Sensitivity	Detection Range	Detection Limit	Response Time	Durability
**TPU/h-BN [7]**	6 × 9	3.997 kPa^−1^	0–100 kPa	4.7 Pa	120 ms	5000 cycles
**ANF/Ti_3_C_2_T_x_ [150]**	4 × 4	521.69 kPa^−1^	0–200 kPa	0.22 Pa	17 ms	10,000 cycles
**TPU/h-BN [154]**	15 × 15	261.4 kPa^−1^	0.05–450 kPa	50 Pa	15 ms	5000 cycles
**PDMS/CNT [155]**	28 × 28	174 kPa^−1^	0.15 Pa–400 kPa	Not known	2.74 ms	10,000 cycles
**PI/Ti_3_C_2_T_x_/Cu [156]**	8 × 8	21.5 kPa^−1^	0.5–410 kPa	140 Pa	60 ms	8000 s
**PDMS/CNT/PEDOT:PSS [157]**	8 × 8	2.32 kPa^−1^	0–10 kPa	Not known	2.5 s	250 cycles
**PBU/AgNW [158]**	10 × 10	888.79 kPa^−1^	1–100 kPa	0.4608 Pa	66 ms	10,000 cycles
**PDMS/PEDOT:PSS [29]**	10 × 1	38.1 Ω/mmHg	0–30 mmHg	Not known	660 ms	200 cycles
**PDMS/Ecoflex/Au [159]**	2 × 5	>5.22 MPa^−1^	45 Pa–4.1 Mpa	Not known	30 ms	5000 cycles
**PDMS/CNT [160]**	8 × 8	0.19 kPa^−1^	0–400 kPa	Not known	80 ms	10,000 cycles
**Ionic gel [161]**	3 × 5	>0.77 kPa^−1^	0–40 kPa	Not known	Not known	Not known
**PDMS/TPU/Conductive ink [162]**	64 × 64	10.50 kPa^−1^	0–130 kPa	1 Pa	69 ms	45,000 cycles

## Data Availability

The data that support the findings of this study are available on request from the corresponding author, upon reasonable request.

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
