# Peer review of "Design, Fabrication, and Application of Large-Area Flexible Pressure and Strain Sensor Arrays: A Review"

_micromachines, 2025, doi:10.3390/mi16030330_

Round 1

Reviewer 1 Report

Comments and Suggestions for Authors

This manuscript titled "Design, fabrication, and application of large-area flexible pressure and strain sensor arrays: A review" by Xikuan Zhang et al. provides a comprehensive overview of the development, fabrication techniques, and applications of large-area flexible sensor arrays. The manuscript is well-structured and offers valuable insights into the field of flexible sensor technology. However, there are a few minor issues that need to be addressed to enhance the clarity and accuracy of the manuscript. Here are the suggested revisions:

1.  Expand the abstract to briefly include the key challenges faced in the fabrication and application of large-area flexible sensor arrays, as well as the potential future directions for research and development.

2.  Include specific examples or case studies where flexible sensor arrays have been successfully implemented in industries such as healthcare, robotics, or wearable technology to illustrate their practical significance. Also, the references to relevant literature in the manuscript are not comprehensive enough. It is recommended to include more related references (eg., 10.34133/research.0157, 10.1002/adfm.202412307, 10.1002/EXP.20230109) to enrich the article.

3. Provide a concise table or a more detailed discussion comparing the fabrication techniques (3D printing, screen printing, laser etching, magnetron sputtering, and molding) in terms of their suitability for different applications, cost-effectiveness, and scalability.

4.  Elaborate on the factors that influence the sensitivity, response time, and stability of resistive and capacitive sensors, such as material properties, structural design, and environmental conditions.

5.  Add a subsection discussing the key factors to consider when selecting materials for flexible sensors, such as flexibility, thermal stability, electrical conductivity, and biocompatibility, with specific examples to illustrate these points.

Reviewer 2 Report

Comments and Suggestions for Authors

This article provides a comprehensive review of the design, fabrication, and application of large-area flexible sensor arrays, emphasizing the critical role of flexible sensors across various fields. The article begins by analyzing the fabrication process, working principles, and commonly used materials for flexible sensors. It then discusses in detail the application of different materials in sensors and describes two traditional fabrication methods. Next, the article systematically reviews the application of large-area flexible sensor arrays in pressure and strain detection. The strength of this work lies in its thorough summary of flexible sensor array technology, with practical applications demonstrated through examples in fields such as intelligent medicine, robotics, and virtual reality, highlighting its broad prospects. However, before this article can be accepted for publication, the author is requested to make minor revisions to enhance the manuscript quality, as outlined below:

  1. It is recommended that the authors revise the English language in the first chapter by correcting the grammatical errors, thereby enhancing the professionalism of the writing.
  2. In Chapter 2, some of the references related to the preparation of flexible sensors are outdated, such as those numbered 79 and 80. It is suggested to incorporate more recent research findings.
  3. In Figure 4, the layout of the images needs to be readjusted to align with the sequence of the text in Chapter 3.
  4. In Chapter 4, please provide the full names of acronyms when they first appear, to ensure that the reader can easily understand the materials being referenced, such as PET, PI, etc.
  5. In the conclusion, the authors are requested to improve the English presentation and provide a more systematic summary of the article's content.

Reviewer 3 Report

Comments and Suggestions for Authors

The manuscript titled " Design, fabrication, and application of large-area flexible pressure and strain sensor arrays: A review" provides a broad review of large-area flexible pressure and strain sensor arrays, covering fabrication techniques, material choices, and applications. While the topic is relevant to Micromachines, the discussion lacks depth, particularly in critical analysis, quantitative comparisons, and scalability challenges. Figures and citations also require improvement. The reviewer recommends minor revisions before reconsideration for publication.

Comment #1. The manuscript summarizes various fabrication techniques but lacks a comparative discussion of their advantages and limitations. Adding a table outlining scalability, cost, sensor resolution, and material compatibility would help highlight the strengths and weaknesses of each method, making the review more informative and balanced.

Comment #2. While different sensor types are discussed, their performance metrics are not well compared. The reviewer suggests including a table summarizing sensitivity, response time, durability, and detection limits, supported by references, to provide a clearer technical comparison of existing approaches.

Comment #3. Some figures lack proper labeling, making it difficult to interpret fabrication steps and sensor structures. The reviewer recommends adding annotations, improving resolution, and merging redundant figures where appropriate to enhance readability and visual effectiveness.

Comment #4. The manuscript does not adequately address challenges in mass production, wireless integration, and real-time data processing for large-area sensor arrays. Discussing manufacturing feasibility and long-term stability would strengthen the review’s practical relevance.

Comment #5. Several statements regarding material properties and applications lack supporting references. The reviewer suggests adding citations to recent experimental studies, particularly from high-impact journals, to ensure claims are well-supported and aligned with the latest advancements.

Comment #6. The manuscript contains grammatical errors, awkward phrasing, and inconsistencies in terminology and formatting. The reviewer recommends a thorough language revision to improve readability, ensure uniform citation formatting, and standardize abbreviations and technical terms throughout the text.

Comments on the Quality of English Language

The manuscript titled " Design, fabrication, and application of large-area flexible pressure and strain sensor arrays: A review" provides a broad review of large-area flexible pressure and strain sensor arrays, covering fabrication techniques, material choices, and applications. While the topic is relevant to Micromachines, the discussion lacks depth, particularly in critical analysis, quantitative comparisons, and scalability challenges. Figures and citations also require improvement. The reviewer recommends minor revisions before reconsideration for publication.

Comment #1. The manuscript summarizes various fabrication techniques but lacks a comparative discussion of their advantages and limitations. Adding a table outlining scalability, cost, sensor resolution, and material compatibility would help highlight the strengths and weaknesses of each method, making the review more informative and balanced.

Comment #2. While different sensor types are discussed, their performance metrics are not well compared. The reviewer suggests including a table summarizing sensitivity, response time, durability, and detection limits, supported by references, to provide a clearer technical comparison of existing approaches.

Comment #3. Some figures lack proper labeling, making it difficult to interpret fabrication steps and sensor structures. The reviewer recommends adding annotations, improving resolution, and merging redundant figures where appropriate to enhance readability and visual effectiveness.

Comment #4. The manuscript does not adequately address challenges in mass production, wireless integration, and real-time data processing for large-area sensor arrays. Discussing manufacturing feasibility and long-term stability would strengthen the review’s practical relevance.

Comment #5. Several statements regarding material properties and applications lack supporting references. The reviewer suggests adding citations to recent experimental studies, particularly from high-impact journals, to ensure claims are well-supported and aligned with the latest advancements.

Comment #6. The manuscript contains grammatical errors, awkward phrasing, and inconsistencies in terminology and formatting. The reviewer recommends a thorough language revision to improve readability, ensure uniform citation formatting, and standardize abbreviations and technical terms throughout the text.
